# Management of possible serious bacterial infection in young infants where referral is not possible in the context of existing health system structure in Ibadan, South-west Nigeria

Adejumoke Idowu Ayede[1,2]*, Oluwakemi Oluwafunmi Ashubu[1,2], Kayode Raphael Fowobaje[1,2], Samira Aboubaker[3], Yasir Bin Nisar[4], Shamim Ahmad Qazi[3], Rajiv Bahl[4], Adegoke Gbadegesin Falade[1,2]

1 Department of Paediatrics, College of Medicine, University of Ibadan and University College Hospital, Ibadan, Nigeria, 2 Centre for African Newborn Health and Nutrition, University College Hospital, Ibadan, Nigeria, 3 Former WHO Staff, Geneva, Switzerland, 4 Department of Maternal, Newborn, Child, Adolescent Health and Aging, World Health Organization, Geneva, Switzerland

* idayede@yahoo.co.uk

## Abstract

## Introduction

Neonatal infections contribute substantially to infant mortality in Nigeria and globally. Management requires hospitalization, which is not accessible to many in low resource settings. World Health Organization developed a guideline to manage possible serious bacterial infection (PSBI) in young infants up to two months of age when a referral is not feasible. We evaluated the feasibility of implementing this guideline to achieve high coverage of treatment.

## Methods

This implementation research was conducted in out-patient settings of eight primary health care centres (PHC) in Lagelu Local Government Area (LGA) of Ibadan, Oyo State, Nigeria. We conducted policy dialogue with the Federal and State officials to adopt the WHO guideline within the existing programme setting and held orientation and sensitization meetings with communities. We established a Technical Support Unit (TSU), built the capacity of health care providers, supervised and mentored them, monitored the quality of services and collected data for management and outcomes of sick young infants with PSBI signs. The Primary Health Care Directorate of the state ministry and the local government led the implementation and provided technical support. The enablers and barriers to implementation were documented.

## Results

From 1 April 2016 to 31 July 2017 we identified 5278 live births and of these, 1214 had a sign of PSBI. Assuming 30% of births were missed due to temporary migration to maternal

**Data Availability Statement:** All relevant data are within the manuscript and its Supporting Information files.

**Funding:** This study was funded by the Bill and Melinda Gates Foundation to the Department of Maternal, Newborn, Child and Adolescent Health, World Health Organization, Geneva, Switzerland and AIA (OPP1114815). The funder had no role in study design, data collection and analysis, decision to publish, or preparation of the manuscript.

**Competing interests:** No competing interest.

homes for delivery care and approximately 45% cases came from outside the catchment area due to free availability of medicines, the treatment coverage was 97.3% (668 cases/ 6861 expected births) with an expected 10% PSBI prevalence within the first 2 months of life. Of 1214 infants with PSBI, 392 (32%) infants 7–59 days had only fast breathing (pneumonia), 338 (27.8%) infants 0–6 days had only fast breathing (severe pneumonia), 462 (38%) presented with signs of clinical severe infection (CSI) and 22 (1.8%) with signs of critical illness. All but two, 7–59 days old infants with pneumonia were treated with oral amoxicillin without a referral; 80% (312/390) adhered to full treatment; 97.7% (381/390) were cured, and no deaths were reported. Referral to the hospital was not accepted by 87.7% (721/822) families of infants presenting with signs of PSBI needing hospitalization (critical illness 5/22; clinical severe infection; 399/462 and severe pneumonia 317/338). They were treated on an outpatient basis with two days of injectable gentamicin and seven days of oral amoxicillin. Among these 81% (584/721) completed treatment; 97% (700/721) were cured, and three deaths were reported (two with critical illness and one with clinical severe infection). We identified health system gaps including lack of staff motivation and work strikes, medicines stockouts, sub-optimal home visits that affected implementation.

## Conclusions

When a referral is not feasible, outpatient treatment for young infants with signs of PSBI is possible within existing programme structures in Nigeria with high coverage and low case fatality. To scale up this intervention successfully, government commitment is needed to strengthen the health system, motivate and train health workers, provide necessary commodities, establish technical support for implementation and strengthen linkages with communities.

## Registration

Trial is registered on Australian New Zealand Clinical Trials Registry (ANZCTR) ACTRN12617001373369.

## Introduction

In 2019, 2.4 million neonatal deaths occurred globally, of which infections contributed to nearly 500,000 deaths [1]. In sub-Saharan Africa, the incidence of possible serious bacterial infection (PSBI) is estimated to be 7.6% with a case fatality of 9.8% [2]. An estimated 10% of live births in low resource settings are expected to develop PSBI [3, 4]. Neonatal mortality remains very high in Nigeria with a rate of 39 per 1000 live births [5]. The recommended treatment for newborns and young infants up to 2 months of age with PSBI is inpatient care with 7–10 days of parenteral antibiotics and other supportive care [6], but for many families, in low resource settings, it is not feasible [7].

Trials in African and Asian countries showed that when a referral is not feasible, simplified antibiotic regimens could be used effectively and safely to treat young infants with PSBI [3, 4, 8, 9]. The evidence generated from these trials contributed to the development of the World Health Organization (WHO) guideline for the management of sick young infants with PSBI when referral is not feasible [10]. Before implementing this guideline at scale, it was deemed

necessary to gain some experience from several countries to demonstrate the feasibility of implementation of this guideline in a programme setting on a limited scale and to assess the acceptability by families.

Thus, the objective of this implementation research in Nigeria was to test the feasibility of using a simplified antibiotic regimen for management of sick young infants with PSBI when a referral was not feasible to achieve high treatment coverage of at least 80%, identify key barriers and learn lessons for potential scale-up. This research was carried out in Zaria, Kaduna Northern part of Nigeria [11] and Ibadan, Oyo State to represent Southern part of Nigeria. We report here data from Ibadan.

## Methods

### Study design

This implementation research was conducted in a programme setting, where we documented processes of implementation in terms of what was done, how it was done, the successes, enabling factors, challenges, barriers and potential solutions.

### Context and study setting

**Study site characteristics.** The study was carried out in Lagelu Local Government Area (LGA), Ibadan, Oyo State, Nigeria. The infant mortality rate is 41 per 1,000 live births and neonatal mortality rate of 30 per 1,000 live births for the State [5]. Proportions receiving antenatal care from a skilled provider, delivered in a health facility and delivered by a skilled provider are 85.4%, 70.1% and 84.6% respectively. Also, the proportion of children aged 12–23 months receiving all basic vaccinations is 23.3% while the percentage for all age-appropriate vaccinations is 16.6% [5].

Ibadan has 11 LGAs, five of which are urban and six are peri-urban and Lagelu is one of the 6 peri-urban LGAs. The estimated population of Lagelu is 150,000 [12]. It has 14 political wards which are the operating health units with each ward having a minimum of one Primary Health Care centre (PHC) or Health Post. Lagelu LGA was selected after extensive deliberations with Federal and Oyo state Ministries of Health (MOH), and other stakeholders because it has rural and peri-urban communities, relatively large size PHCs, Community Resource Persons (CORPs) are available and LGA administrators' were ready to support the implementation. After further consultation with the health directorate, four wards (1, 4, 5 and 7) were selected, which were typical of rural communities where referral may not be feasible and had eight PHCs. The rural population are mainly Yoruba peasant farmers, petty traders and artisans with low income. Literacy level is low with the majority having a 6- years of education. Polygamous settings are common and most mothers have their health decisions influenced by husbands, co-wives, religious and traditional leaders and mothers- in-laws all playing the role of gate-keepers and significant others [13].

**Health system structure.** Three levels of health structure operate in Nigeria; primary, secondary and tertiary and all three are available in Ibadan where sick infants from the selected LGA and wards are seen. The primary level is PHC under the health directorate of Local Government and State Ministry of Health, the smallest health care delivery unit serving a population of 5,000–6,000. It is located 1–2 kilometres from the communities. The services rendered are mainly outpatient with minimal or no admission facilities and reach the people through ward development health committees. The main role of PHC is to offer preventive and curative services. The PHCs are manned by doctors when available, nurses/midwives, Community Health Officers (CHOs) and Community Health Extension Workers (CHEWs). CORPs are used in community-based programmes dealing with maternal and child health, malarial drugs

and insecticide-treated nets (ITN) distribution and are paid monthly stipend. CORPs minimum level of education is secondary school and most have been trained in integrated community case management (iCCM) [14] and home-based care of newborn. iCCM training materials were developed by WHO and UNICEF for the community level health workers to identify and refer common illnesses in under five-year-old children. Referral from PHC is to a secondary level general hospital, which is staffed with medical doctors and nurses. In some general hospitals, there may be specialist doctors trained in paediatrics or neonatology and may have a newborn nursery. Referral may also be to tertiary centres with specialized care. The referral centres for the LGA have newborn in-patient admission services including oxygen supply, parenteral antibiotics, and respiratory support but no family centred care.

**Study population.** Young infants aged up to 2 months who had signs of PSBI identified at home and referred to PHC by CORPs or brought directly to the PHC by the caregivers.

## Implementation of the study

The implementation of this study was based on the RE-AIM (Reach, Efficacy/Effectiveness, Adoption, Implementation, Maintenance) framework [15] as shown in Fig 1.

**Policy dialogue at national, state and local level.** In March 2015, a two-day orientation and policy dialogue meeting was held in Abuja, organised by the Federal MOH in collaboration with the WHO. Participants included senior federal and state MOH policymakers and programme implementers, representatives from the professional associations, stakeholders at national, sub-national levels and development partners, as well as WHO and UNICEF. The status of newborn and child health situation in Nigeria, programme strategies, evidence from the trials [3, 4, 8, 9] and WHO PSBI guideline [10] were presented and discussed. Meeting participants reviewed the research findings and discussed policy implications related to the implementation of the WHO PSBI guideline. It was decided to create a platform for learning through implementation research; to learn how the WHO PSBI guideline for managing young infants with PSBI when referral is not feasible can be implemented in a programme setting in Nigeria. It was agreed to set up two implementation research sites, one in Zaria, Kaduna State, Northern Nigeria [11] and the other in Ibadan, Oyo State, Southern Nigeria. Boxs 1 and 2 summarize the issues that were discussed and decisions made.

Following the national and state-level dialogue, meetings with the Lagelu LGA chairman, senior administrators and the PHC coordinator were also held at their secretariat.

**Community sensitization.** Several meetings were organized at the early phase with community leaders–local chiefs, religious leaders, market women and men associations, traditional birth attendants (TBA), and focal persons such as ward leaders and landlord associations. The number of such meetings varied from 3–5 per community served by the PHC and they were informed about the research and its potential benefits to the communities. These community leaders assisted in the selection of the CORPs using the following criteria: resident in the community, are available and can read and write with the minimum educational level of secondary school. These meetings ensured community acceptability of home visit in pregnancy and post-delivery follow up and their continuous involvement in the implementation of the study in their respective communities.

**Establishment of a Technical Support Unit (TSU).** To kick start implementation, a TSU was set up at the University College Hospital, Ibadan and funded by the project. Members included the Principal Investigator, Project Manager, two Project Supervisors, eight Project Nurses, Executive Secretary of Oyo State Primary Health Care Board and a representative of the Local Government Health Directorate. All were master trainers for Integrated Management of Childhood Illnesses (IMCI), iCCM and training on Caring for the Newborn at Home

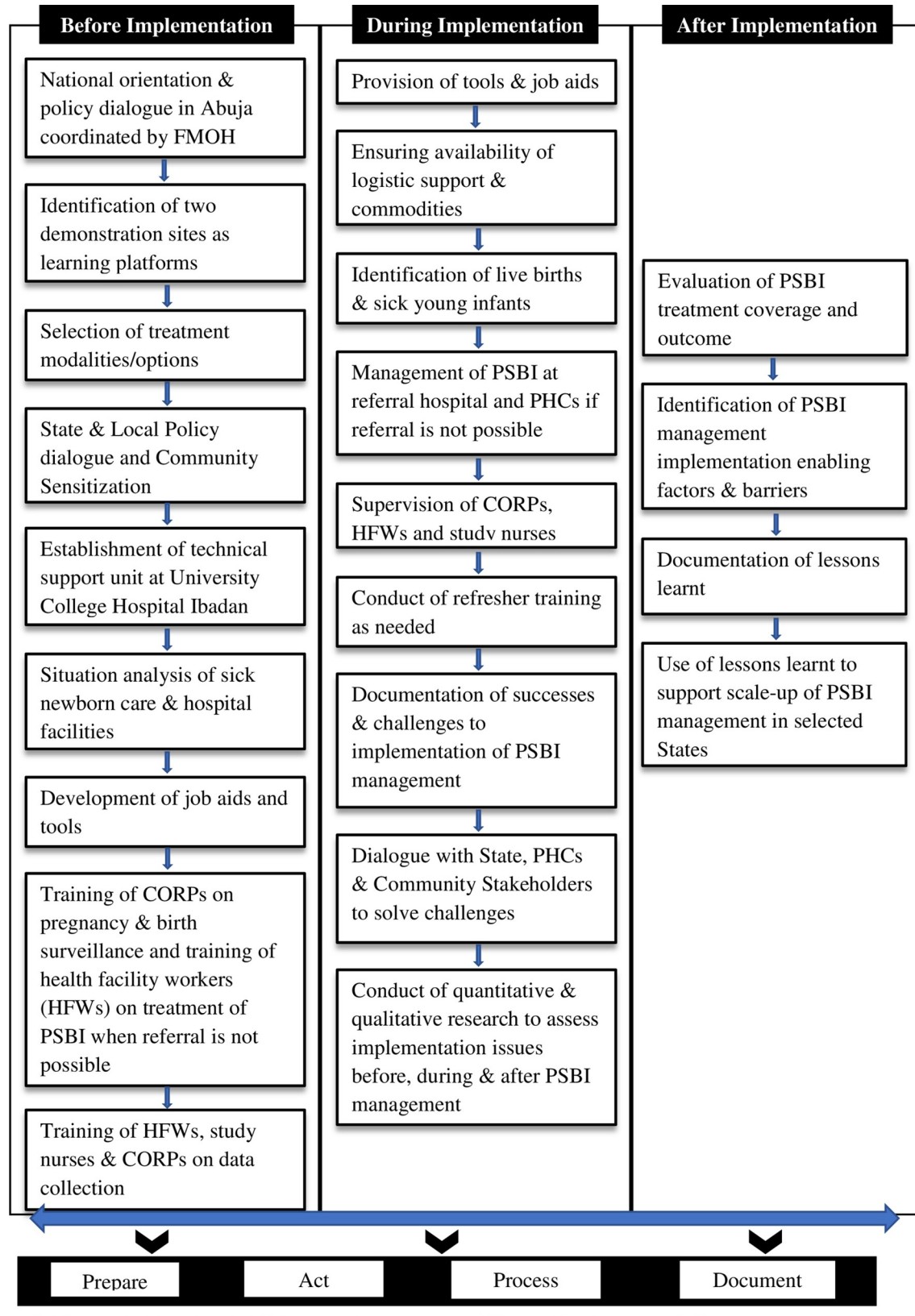

FMOH: Federal Ministry of Health, CORPs: Community Resource Persons, HFWs: Health Facility Workers, PSBI: Possible Serious Bacterial Infection, PHCs: Primary Health Centres.

**Fig 1. Conceptual framework for the implementation of the study.**

Box 1. Definitions and treatment of PSBI [10]

**Definition:** PSBI is defined as young infant 0–59 days presenting with any of the following signs: fast breathing (respiratory rate ≥60 per minute), severe chest indrawing, no movement at all or movement only when stimulated, not able to feed at all or not feeding well/stopped feeding well, convulsions, high body temperature (≥ 38˚ C) and low body temperature (<35.5˚ C)

**Classification:**

• **Critical Illness (CI)**–infant 0–59 days presenting with one or more of the following signs: convulsions, not able to feed at all or no movement on stimulation

Treatment: Refer urgently to a hospital. If the referral is not possible after multiple attempts treat with daily injectable gentamicin and twice-daily ampicillin until a referral is possible or for at least 7 days. Daily follow-up till referral is feasible.

• **Clinical severe infection (CSI)**–infant 0–59 days presenting with one or more of the following signs: not feeding well, high body temperature (38˚C or above), low body temperature (less than 35.5˚C), severe chest indrawing or movement only when stimulated.

Treatment: Refer to a hospital. If the referral is not feasible, treat with injectable gentamicin daily for 2 days and 7 days oral amoxicillin. Counsel family on each occasion. Mandatory follow-up on day 4 of treatment.

• **Severe pneumonia**–only fast breathing (60 breaths per minute or more) in infants 0–6 days of age

Treatment: Refer to a hospital. If the referral is not feasible, treat with oral amoxicillin for 7 days. Mandatory follow-up on day 4 of treatment.

• **Pneumonia**–infant 7–59 days presenting with only fast breathing (60 or more breaths per minute)

Treatment–oral amoxicillin for 7 days without a referral. Mandatory follow up on day 4 of treatment.

[14, 16, 17]. The IMCI training materials were developed by WHO and UNICEF to manage common childhood illnesses for professional health workers working at first-level facilities or outpatient departments. This TSU was set up for the project and their funding ended with the project but the staff was available to share experiences and provide technical support for scale-up in other states. The role of the TSU is described in Box 3 below.

**Preparations for the implementation of PSBI guideline.** Consultations were held between TSU staff and State MOH and PHC directorate in the public health system. Agreements were made between the two for collaboration and delineation of various tasks. The PHC directorate was to take part in facilitating the research, creating an enabling environment for health workers, ensuring routine and essential newborn care services and timely supply of commodities. They were also to promote and support post-natal home visits by CORPs, home visits by CORPs for infants who failed to present for follow up, ensure prompt and accurate record-keeping and facilitate the referral. The clinical staff [nurses, CHEWs, CHO] of the PHCs were to identify and classify sick young infants, refer to a hospital who needed it, reclassify if the referral was not feasible, treat and follow up those who were treated on an outpatient basis. The level of training, qualifications and responsibilities of these PHC health workers are described by Wammanda et al [11].

The potential cost for the involvement of CORPs, ensuring regular supply of drugs and commodities, purchase of medicines were also deliberated upon. Payment of stipend of 10,000 Nigerian Naira (approximately 33 US$) by the project fund to each CORP and bulk purchase of consumables especially medicines from local manufacturers were identified as means of reducing cost and ensuring a smooth implementation.

**Situation analysis of sick newborn care and health facilities.** A situation analysis of the health facilities was conducted before the implementation of PSBI activities, which included a review of the availability of staff and their capacity at each PHC (S1 Table), facility leadership, supply chain, consumables inflow and outflow, sharps disposal and sick newborn registers. In general, all sick newborns and young infants were assessed at a PHC and referred from the LGA to either a secondary health facility, Oni Memorial Children's Hospital; or tertiary facility, University College Hospital and Adeoyo Maternity Teaching Hospital in Ibadan city. The

## Box 2. Policy dialogue decisions and recommendations

| Issues | Decision |
|---|---|
| • Who will identify sick young infants in the community? | • State-based Community Resource Persons (CORPs)/ Village Health Workers (VHWs)/Community Health Extension Workers (CHEWs) and not Patent and Proprietary Medicine Vendors (PPMVs) |
| • Where will the sick young infants with PSBI be referred from the community? | • Primary Health Care Centres (PHCs) |
| • Who will confirm pneumonia, severe pneumonia, clinical severe infection and critical illness and refer those who need it to the hospital? | • CHEWs/Nurses/Midwives/Community Health Officers (CHO), who so ever is available at the PHC. |
| • Who will treat if the family does not accept a referral to a hospital? | • CHEWs/Nurses/Midwives/Community Health Officers (CHO), who so ever is available at the PHC |
| • Where will the treatment be provided? | • PHC on an outpatient basis |
| • What antibiotic therapy will be used for the treatment of clinical severe infection at PHC if a referral to the hospital is not accepted? | • Intramuscular gentamicin 5–7.5 mg/kg (for low-birth-weight infants gentamicin 3–4 mg/kg) once daily for two days and twice-daily oral amoxicillin, 50 mg/kg per dose for 7 days |
| • Which antibiotic will be used for severe pneumonia in infants 0–6 days of age when a referral is not feasible? | • Oral amoxicillin 50 mg/kg per dose twice daily for 7 days |
| • Which antibiotic will be used for pneumonia in infants 7–59 days of age without a referral? | • Oral amoxicillin 50 mg/kg per dose twice daily for 7 days |
| • Where will follow up be done? | • PHCs |
| • Where will implementation research sites be? | • Ibadan in Oyo State and Zaria in Kaduna State |

Abbreviations: CHEWs; community health extension workers, CHO; community health officers, CORPs; community-oriented resource persons, PHC; primary health care centre, PPMV; Patent and Proprietary Medicine Vendors; (private individuals operating a chemist shop who are not qualified pharmacists), VHW; village health workers.

distance between the LGA and these facilities ranges between 10–40 km and the average cost of transportation is 600 Nigerian Naira (approximately 2 US$). At the PHC and these referral hospitals, cost of treatment is mainly out of pocket and occasionally medicines are subsidized through a revolving fund [18]. In the case of stockouts, patients had to buy medicines. Routine follow-up was not done to find out if a referral to a hospital was accepted or not. This management was in line with the iCCM or IMCI [14, 16, 17] strategies introduced and implemented in Nigeria.

**Training of health workers.** Master Trainers were trained by WHO facilitators. TSU in collaboration with the LGA organized training for nurses and CORPS. CORPs received

## Box 3. Role of the technical support unit

• Provide technical assistance in the implementation of PSBI guideline by the PHCs when a referral is not feasible.
• Support development and adaptation of training materials and job aids.
• Build capacity for case management through hands-on clinical training of health workers.
• Supervision and mentoring.
• Monitor quality of services including follow up.
• Ensure availability of commodities.
• Document implementation and use findings to facilitate policy dialogue and problem-solving.
• Data collection.

training on 'Caring for the newborn at home', a WHO/UNICEF training course for community health workers [17]. The study nurses, health facility nurses and CHEWs needed to have a broader understanding and hence were trained on 'Caring for the newborn at home' and the 'Management of sick young infants aged up to 2 months', a component of IMCI, WHO/UNICEF training module [19]. The 6 days training included theoretical knowledge, skills development using clinical practice at Oyo State Oni Memorial Children's Hospital, games, role plays, interactive communication and group discussions. They were also trained on data collection using different case records forms. Multimedia, flip charts and training manuals were used. During and post-training skill acquisition was assessed through standardization exercises and field practice which included supervised hospital and home visits.

**Tools and job aids.** The TSU adapted and updated tools and jobs aids including case record forms (CRFs), counselling cards, mother and baby card, referral notes, young infants IMCI Chart booklet and calendars for easy identification of dates for follow up. These were provided to the workers. When required health workers were provided with respiratory rate timers, digital thermometers, weighing scales, backpacks, mobile phones and umbrellas through project funds.

**Logistics and commodities.** Oral amoxicillin dispersible tablets (DT), injectable gentamicin, 2 ml and 5 ml needles and syringes, cotton wool, methylated spirit and safety box were purchased in bulk locally through the project funds as these were not routinely supplied by the State and LGA. They were provided free of charge to the sick young infants with PSBI and the project was also responsible for cost of home visit.

**Identification and management of sick young infants with PSBI.** *Identification of live births.* CORPs identified pregnancies and births through house-to-house visits, communication with pregnant mothers (who assisted in identifying other pregnant mothers), TBAs, health centres staff, community leaders and community spokespersons.

*Identification of sick young infants and referral to PHC.* CORPs made post-natal home visits on days 1, 2, 3, 7 and 14 days of birth to identify sick infants who needed to be referred to the PHC. Mothers were also counselled on signs of PSBI and prompt appropriate care-seeking.

*Management of PSBI either at PHC or referral level facility.* Each PHC had a trained HFW who examined the sick infant referred by the CORP or brought by the mother, sometimes assisted by the study nurse. Infants with signs of PSBI confirmed by HFW needing referral to a hospital were referred for appropriate treatment. However, when a referral was not feasible, further assessment and re-classification of the sick young infant were done by the HFW using the WHO young infant IMCI chart booklet [19] as a guide and prompt management was started according to the agreed protocol (Box 2). Mandatory follow up of sick infants on treatment was done at the PHC as described in Box 1 but those that did not come for follow up were visited at home by CORPs and study nurses. Treatment outcome of all sick infants was evaluated on day 14 of enrolment by study nurses while data on treatment adherence and follow up were collected by HFW and assisted by the study nurses.

**Monitoring, supervision, skills maintenance and process evaluation.** To ensure quality, study supervisors performed routine visits starting with once a week per facility for health facility worker (HFW), three times a week per facility for the study nurses and daily for CORPs for direct observations using a monitoring form and a supervisory checklist (S1 File). Visits to HFWs included observation of assessment, classification and treatment of sick infants, retention of trained HFW, checking for implementation challenges and availability of commodities and job aids. Visits to study nurses included observing and checking the filling of IMCI assessment form and treatment form. Recording forms kept by CORPs were reviewed for correct filling of CRFs (pregnancy register and postnatal visit) and strict adherence to follow-up dates using the study calendar. The investigators also conducted random field visits to

ensure internal quality control. Refresher training courses of various workers were conducted quarterly to ensure skills maintenance and quality standards. In-depth interview and focussed group discussions were conducted with mothers of sick young infants and Key Informant interview was conducted with the HFWs using standard interview guide by trained experts in qualitative research details of which will be published separately.

### Data management and analysis

All completed CRFs from the field were received by the data nurse/ supervisor every month. These forms were checked for completion and errors, corrected after queries were resolved and were handed over to the data manager. Epi Data version 3 was used for dual data entry. Data validation and verification were performed by the data manager, after which data was stored on specially designated computer and backed up on an external drive. Descriptive analysis was conducted to compute frequencies and proportions for the various variables including the proportion of sick young infants identified with PSBI and different sub-classifications, proportion accepted or refused referral, proportion completed treatment, follow-up and their outcomes using Stata statistical package version 13.

We collected data on barriers and enabling factors for implementation by direct observation, documentation of events and interaction with PHC staff. Findings were regularly shared with programme implementers and local health authorities through stakeholders meetings, during joint visits by the PHC Board Executive Secretary and in monthly meetings with heads of PHCs. The barriers were resolved through round table discussion with heads of PHCs, PHC Health Directorate at State and LGA level. Critical appraisal of the causes and factors resulting in challenges and potential solutions were discussed.

**Consent and ethical approval.** The implementation research was approved by the Institutional Review Board of the University of Ibadan UI/EC/15/0177 and by the WHO Ethical Review Committee. Written informed consent was obtained from all eligible caregivers prior to their participation in the study.

## Results

We identified 5278 live births over a period of 16 months from 1 April 2016 to 31 July 2017 when the implementation research was carried out. The newborns followed up by CORPS at home on days 1, 3, 7 and 14 were 60.6% (3198/5278), 71.2% (3756/5278), 79.7% (4209/5278) and 96% (5069/5278), respectively.

### Identification of infants with PSBI and referral

Out of the 5278 live births identified at different days of life, and followed up, 1214 were confirmed with signs of PSBI at PHCs. The highest number of young infants with PSBI were seen in the second 4 month period (Fig 2). Assuming 30% births were missed (similar to estimates from Zaria [11] and Ethiopia [20] due to temporary migration to maternal homes for delivery care and approximately 45% cases came from outside the catchment area (obtained from home visit address register) due to free availability of medicines, the treatment coverage was 97.3% (668 cases/6861 expected births) with an expected 10% PSBI incidence within first 2 months of life [3, 4].

CORPs identified 1133 (93.3%) infants in the community and referred them to PHCs. Among 1214 infants with any sign of PSBI at PHCs, 392 (32.3%) infants 7–59 days old had pneumonia, 338 (27.8%) infants 0–6 days old had severe pneumonia, 462 (38.1%) had clinical severe infection, while 22 (1.8%) had critical illness (Table 1).

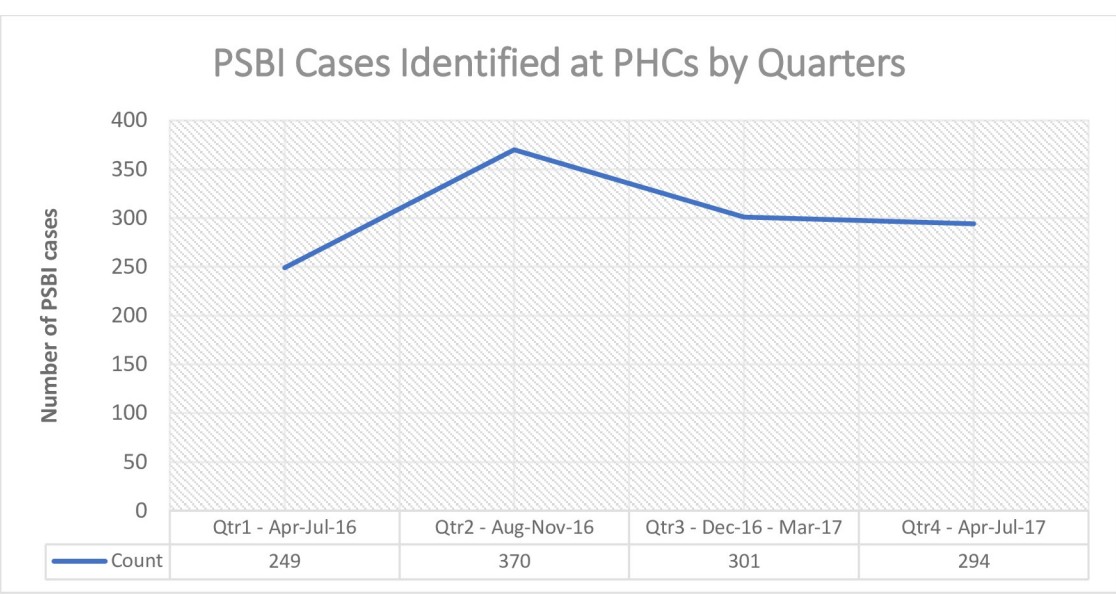

**Fig 2. Identifaction of PSBI cases at the PHCs by quarters.**

## Treatment and outcome of treatment

Among 7–59 days infants with pneumonia, 99.5% (390/392) received outpatient treatment at the PHC with 7 days oral amoxicillin without a referral; 80% (312/390) completed treatment, 79% (308/390) completed all follow-ups, 96% (378/390) were cured, 2.3% (9/390) failed therapy with no deaths (Tables 1 and 2). Among 0–6 days infants with severe pneumonia, 93.8% (317/338) received outpatient treatment at the PHC with 7 days oral amoxicillin after they refused a referral to the hospital; 81.7% (259/317) completed treatment, 79.2% (251/317) completed all follow-ups, 96.5% (306/317) were cured with no deaths (Tables 1 and 2).

Among 462 infants with signs of clinical severe infection, 86% (399/462) refused a referral to a hospital and agreed to receive outpatient treatment at the PHC (2 days gentamicin and 7 days oral amoxicillin); 81.2% (324/399) completed treatment, 80.7% (322/399) completed all follow-ups, 98% (391/399) were cured, 1.8 (7/399) failed therapy with one death (Tables 1 and 2). Of the 22 infants with critical illness, referral to the hospital was not accepted by 5 (23%), who were treated on an outpatient basis at the PHC; 3 failed treatment with 2 deaths (Tables 1 and 2).

Overall, only 8.5% (103/1214) accepted referral to a hospital and the case fatality rate was 0.5% (3/1214).

## Challenges and successes during the implementation of management of PSBI in PHC

Throughout the implementation process, we documented the challenges related to the health system and implementation issues and processes to resolve them (Table 3). The success recorded in this implementation research was due to a lot of factors related to appropriate communication with policymakers and major stakeholders, integration of intervention into the health system structure, ensuring the availability of commodities and medicines, training of all staff involved, making job aids available and collaboration in solving problems. The role of the TSU in providing training, monitoring and supervision improved the overall quality of services rendered. The early identification of challenges through timely monitoring and

**Table 1. Identification, treatment adherence and follow-up of young infants 0–59 days old with signs of PSBI (n = 1214).**

| Parameters | 7–59 days Fast breathing* only n/N (%) | 0–6 days Fast breathing* only n/N (%) | 0–59 days signs of Clinical severe infection† n/N (%) | 0–59 days signs of Critical illness‡ n/N (%) |
|---|---|---|---|---|
| Infants identified at the primary health care centers (PHC) | 392/1214 (32.3) | 338/1214 (27.4) | 462/1214 (38.1) | 22/1214 (1.8) |
| Infants brought directly by families to a PHC | 15/392 (3.8) | 8/338 (2.4) | 53/462 (11.5) | 5/22 (22.7) |
| Infants identified by CORPs in the communities and referred to a PHC | 377/392 (96.2) | 330/338 (97.6) | 409/462 (88.5) | 17/22 (77.3) |
| Infants referred to the hospital from a PHC | 2/392 (0.5) | 338/338 (100) | 462/462 (100) | 22/22 (100) |
| Infants whose parents/family refused a referral to the hospital from PHC | Not applicable | 317/338 (93.8) | 399/462 (86.1) | 5/22 (22.7) |
| Infants whose parents/family accepted treatment on an outpatient basis at a PHC | 390/392 (99.5) | 317/338 (93.8) | 399/462 (86.1) | 5/22 (22.7) |
| **Adherence and follow-up of patients treated on an outpatient basis** | | | | |
| **Adherence to treatment** | | | | |
| Infants who completed full treatment§ | 312/390 (80.0) | 259/317 (81.7) | 324/399 (81.2) | 1/5 (20.0) |
| Infants who received two injections of gentamicin | Not applicable | Not applicable | 392/399 (98.2) | 0/5 (0.0) |
| Infants who received one injection of gentamicin | Not applicable | Not applicable | 6/399 (1.5) | 2/5 (40.0) |
| Infants who did not receive any injection of gentamicin | Not applicable | Not applicable | 1/399 (0.3) | 2/5 (40.0) |
| Infants who received all 14 doses of oral amoxicillin | 312/390 (80.0) | 259/317 (81.7) | 324/399 (81.2) | Not applicable |
| Infants who received 10–13 doses of oral amoxicillin | 9/390 (2.3) | 18/317 (5.7) | 19/399 (4.7) | Not applicable |
| Infants who received 6–9 doses of oral amoxicillin | 24/390 (6.2) | 19/317 (6.0) | 38/399 (9.5) | Not applicable |
| Infants who received 5 or fewer doses of oral amoxicillin | 42/390 (10.8) | 21/317 (6.6) | 18/399 (4.5) | Not applicable |
| Missing information | 3/390 (0.8) | 0/317 (0.0) | 0/399 (0.0) | 0/5 (0.0) |
| **Follow up** | | | | |
| Infants who were visited on all follow ups (days 1–7 and 14) | 308/390 (79.0) | 251/317 (79.2) | 322/399 (80.7) | 2/5 (40.0) |
| Infants who were visited on day 4 mandatory follow up | 375/390 (96.2) | 307/317 (96.8) | 388/399 (97.2) | 2/5 (40.0) |
| Infants who were partially followed-up (all follow-up visits not completed) | 79/390 (20.2) | 66/317 (20.8) | 77/399 (19.3) | 3/5 (60.0) |
| Infants who were declared lost to follow-up | 3/390 (0.8) | 0/317 (0.0) | 0/399 (0.0) | 0/5 (0.0) |

*Fast breathing defined as the respiratory rate of 60 or more breaths per minute.

†Clinical severe infection defined as the presence of any one of the following signs: i) severe chest indrawing; ii) high body temperature ($\geq 38°C$); iii) low body temperature ($< 35.5°C$); iv) movement only when stimulated; v) stopped feeding well.

‡Critical illness defined as any one of the following signs: i) not able to feed at all; ii) convulsions or fits; iii) no movement at all.

When referred by a CORP in the community to a primary health care center, 35 families with young infants 7–59 days with fast breathing, 37 families with young infants 0–6 days old with fast breathing, 47 infants 0–59 days old with signs of clinical severe infection and 5 infants with signs of critical illness refused CORP's referral advice.

§ For young infants 7–59 days with fast breathing and young infants 0–6 days old with fast breathing both it was only 14 doses of oral amoxicillin; for 0–59 days old with signs of clinical severe infection it was two injections of gentamicin plus 7 days of oral amoxicillin, and for infants with signs of critical illness it was injection gentamicin for 7 days once daily and twice-daily injection of ampicillin for 7 days.

supervision by the TSU and periodic meetings with the health managers provided opportunities to resolve them. Extensive community engagement led to the acceptability of the services rendered at the PHC, increase in care-seeking and follow-up of sick infants.

## Lessons learnt and scale-up support for other stakeholders

There were several lessons learnt from the various phases of the implementation. Extensive policy dialogue at all levels–National, State and LGA played a significant role in ensuring a smooth implementation. CORPs used for pregnancy and birth surveillance are readily

**Table 2. Place of treatment and outcome for young infants with signs of PSBI (N = 1214).**

| PSBI Classification | Hospital Treatment | | Outpatient Treatment | | |
|---|---|---|---|---|---|
| | Number Treated | Deaths within 15 days | Number Treated | Clinical Treatment Failure Excluding deaths** | Deaths within 15 days of treatment |
| 7–59 days fast breathing only n = 392 (%)* | 2 (0.5) | 0 (0.0) | 390 (99.5) | 9 (2.3) | 0 (0.0) |
| 0–6 days fast breathing only n = 338 (%)* | 21 (6.2) | 0 (0.0) | 317 (93.8) | 11 (3.5) | 0 (0.0) |
| 0–59 days Clinical severe infection n = 462 (%)† | 63 (13.6) | 6 (9.5) | 399 (86.4) | 7 (1.8) | 1 (0.3) |
| 0–59 days Critical illness n = 22 (%)‡ | 17 (77.3) | 1 (5.9) | 5 (22.7) | 3 (60.0) | 2 (40.0) |
| Total PSBI n = 1214 (%) | 103 (8.5) | 7 (6.8) | 1111 (91.5) | 30 (2.7) | 3 (0.3) |

*Fast breathing defined as the respiratory rate of 60 or more breaths per minute.

†Clinical severe infection defined as the presence of any one of the following signs: i) severe chest indrawing; ii) high body temperature ($\geq 38°C$); iii) low body temperature ($< 35.5°C$); iv) movement only when stimulated; v) stopped feeding well.

‡Critical illness defined as any one of the following signs: i) not able to feed at all; ii) convulsions or fits; iii) no movement at all.

**Treatment failure in young infants 7–59 days with Fast breathing pneumonia group is defined as:**

• The appearance of new sign of 'critical illness (CI)' or 'clinical severe infection (CSI)' up to day 8 of treatment (worsening). **OR**

• Persistence of fast breathing (Respiratory rate 60 breaths or more per minute) on day 8 of treatment (will be referred for further evaluation–no new treatment to be started at an outpatient level).

**Treatment failure in young infants <2 months of age with CI/CSI/fast breathing up to 6 days is defined as:**

• The appearance of any new sign of CI or CSI up to day 8 of treatment (worsening). **OR**

• Persistence of any presenting sign of CSI or CI on day 4. **OR**

• In CSI cases persistence of any presenting sign by day 8 of treatment. **OR**

In 0–6 days old FB group only, the persistence of fast breathing on day 8 (will be referred for further evaluation).

available in the community but presently no sustainable budget line is available for remuneration by the State/LGA. Treatment of PSBI can be successful if HFWs are adequately trained, mentored and well-motivated, and commodities and job aids are made available to them. The drugs and commodities for the treatment of PSBI are locally available but are not regularly supplied through State/LGA system. Insufficient number of HFWs in the PHCs, irregular payment of salaries result in low motivation. Lack of support for a home visit is a major challenge for follow-up activities needed for successful home-based newborn care.

The TSUs staff from Ibadan and Zaria sites shared preliminary results and initiated discussions on policy options with national and state stakeholders twice, once in 2016 and once in 2017. This served as a platform to address some concerns regarding treatment options, management of sick young infants in PHC, CHEWs giving injections, follow up issues and possible increased risk of antibiotic resistance. The FMOH senior staff visited the implementing PHCs to gain first-hand knowledge of the progress, barriers and facilitators. Training courses were held by TSU trainers for the staff of three international non-governmental organizations (NGOs), Maternal and Child Support Programme (MCSP) Nigeria/Save the Children, USAID and PACT Nigeria, who also visited the implementing PHCs. Additional training courses for Master Trainers were held in Oyo, Ebonyi, Kogi and Niger States with their support. Furthermore, the lessons learnt were used in incorporating PSBI management in young infants when a referral is not possible into the Nigerian Essential Newborn Care Course (ENCC) as part of IMCI and in the iCCM training modules. Technical support was provided to train all Nigerian ENCC National Course Directors on management of PSBI in young infants when referral is not possible.

**Table 3. Challenges to implementation and solutions.**

| Challenges | Actions taken | Outcome |
|---|---|---|
| Insufficient number of staff working in the primary health care centres (PHC) | This challenge was extensively discussed with the PHC Board, Permanent Secretary of State Ministry of Health and Ministry of LGA who made promises to employ more staff | It was not resolved till the end of the implementation research project period and the PHC staff had to be supported by the TSU nurses in some cases |
| Time needed to assess sick young using WHO IMCI chart booklet | This was resolved by printing a single wall chart in large font describing assessment, reclassification and treatment. It was put on the wall beside the consulting table for easy visualization without needing to flip through the chart booklet. A nurse hired by TSU to support 1 to 2 PHCs provided technical assistance to the PHC staff through on the job training, demonstration and acquisition of skills and collected research-related data. | This problem was resolved by the third month of the project implementation |
| Difficulties associated with follow up of sick infants on treatment | It was resolved through the use of community, religious and traditional leaders with the support of TSU having meetings with the members of the communities emphasizing the need to ensure early identification, prompt and adequate treatment for the survival of their sick infants. | Resolved within the first three months of the project. |
| Lack of commodities and job aids | Commodities such as cotton wool, spirit, needles and syringes and drugs; injection gentamicin, oral amoxicillin dispersible tablets and injection ampicillin were supplied by the project. Job aids such as weighing scales, thermometers and respiratory timers were also provided by the project as they were not available at the PHCs. | This problem persisted throughout the study period despite repeated discussions with the government at the local and state level |
| Reduced staff motivation and staff going on strike | During this period of staff strikes due to severe delay in payment of salaries, the sick infants were seen at home by TSU nurses. CORPs continued to conduct pregnancy and birth surveillance as well as identification of ill infants with PSBI. The issue of frequent transfer of trained staff in PSBI management was resolved by identifying a senior nurse in each facility who could transfer the skills easily and where this was not possible, we had retraining sessions. | The strike period lasted three months during the project implementation The problems associated with staff transfer were resolved within the first four months of project implementation. |

## Discussion

We demonstrated that it was feasible to implement the WHO PSBI guideline within the existing programme setting in Ibadan, Nigeria. We were able to identify nearly all the expected sick young infants with PSBI with the number increasing with time as a result of improved community awareness and acceptability. We provided infants with treatment achieving coverage of 97%, with low treatment failure and case fatality rates. We established that 7–59 days old infants with pneumonia can be successfully treated with oral amoxicillin on an outpatient basis without a referral, making this an important intervention for increasing access to treatment.

In our setting, most families (88%) did not accept a referral to a hospital as had been reported by earlier trials [3, 4, 8, 9]. The acceptance of referral advice varied by infection severity. In our study, 77% of infants with critical illness accepted the referral, while only 14% accepted referral advice for infants with clinical severe infection. Similar issues regarding referral were also observed in the other implementation research studies that evaluated the feasibility of WHO PSBI guideline in their settings. In Zaria, Nigeria 97% of families refused referral [11], in Malawi over 90% of infants with signs of critical illness or CSI or severe pneumonia refused referred to a district hospital [21]. Similarly, in Lucknow, India, 81% refused referral advice to a hospital [22] and in MaMoni project, Bangladesh 71% of the families refused referral advice [23]. All were treated on an outpatient basis at health centres. In contrast, in 90% of those who needed referral accepted it and were treated at a hospital in Pune, India [24]. Common reasons that were identified as barriers to accepting referral advice were economic, distance to a hospital, family or cultural reasons, perception of low quality of care at the hospitals [11, 21, 22, 25].

We compared our treatment and outcome data with other PSBI implementation research carried out in various countries. Our high coverage of treatment was similar to Zaria, Nigeria (96%) [11] but was higher than that reported from Malawi (64%) [21], Lucknow, India (53%) [22]; Pune, India (57%) [24] and Kushtia, Bangladesh (31%) [26]. Our over 80% treatment completion rates were similar to MaMoni, Bangladesh (80%) [23] and lower than Malawi (95%) [21] and Zaria, Nigeria (94%) [11]. Our treatment failure rates for infants with signs of CSI treated on an outpatient basis was 2%, compared to 1.3% in Zaria, Nigeria [11]; 3.5% in Malawi [21], and 5.8% in Lucknow, India [22]. For pneumonia in 7–59 days old infants, our treatment failure rate was 2.3%, compared to none in Zaria, Nigeria [11], 2.2% in Malawi [21] and 4.8% in Lucknow, India [22]. Our overall case fatality rate was 0.8% was a bit higher than Malawi (0.2%) [21], but lower than that reported from Lucknow, India (2.6%) [22], Zaria, Nigeria (3.6%) [11] and Pune, India (3.4%) [24]. However, the case fatality for critical illness treated at out-patient level was very high (40%), which supports the recommendation of inpatient treatment for such patients [6]. Overall, low case fatality rates from sites that implemented WHO guideline when the referral was not feasible were much lower than 9.8% reported in a systematic review from low and middle-income countries [2]. We believe the contributory factors to achieve this low case fatality were early identification, appropriate care-seeking, treatment and follow-up of these sick young infants near their homes on outpatient basis by trained and well supervised health workers when referral was not feasible. In addition, there was high acceptability of outpatient treatment by the families in this situation.

Post-natal home visits by CORPS in our setting helped identify most sick young infants. In our case, the proportion of newborns who received home visits increased from 71% in the first 3 days to 96% by day 14. The role of supervisors and community engagement in ensuring proper and timely home visits could have been responsible for this high coverage. It was much higher than what has been reported from large scale programmes i.e., 57% in Bangladesh, 11% in Malawi, and 50% in Nepal [27]. Community health workers (CHWs) play a critical role in the identification and management of infants with signs of PSBI. Community-based studies in Ethiopia [28], and Nepal [29] demonstrated the contributions of CHWs in the identification and treatment of infants with PSBI. In Lucknow, India 81% of young infants with PSBI were identified by Accredited Social Health Activists [22] and 89% were identified by CORPs in Zaria, Nigeria [11] comparable to ours of over 80%.

We have learned several lessons from this early implementation of WHO PSBI guideline in a programme setting. The high treatment success rate with low case fatality rates indicate that this intervention has contributed to saving many infant lives in Ibadan, Nigeria making it a safe and effective strategy for policymakers. However, if the Government of Nigeria wants to scale up this intervention, several aspects would need to be considered to make it a success. It takes some time for the intervention to be established well, and it took us over six months before all essential components were functioning well. Health system support is needed in various places. The health workers need to be motivated by ensuring regular payment of their salaries, building their capacity, providing job aids, supportive supervision and mentoring. Refresher or on the job training to maintain their skills, merit awards to the hard-working staff and a path to promotion would improve the morale of the health workers. Where needed adequate staff should be provided to reduce workload. Monitoring and supervision should be supported through the LGA maternal, newborn and child health unit of PHC directorate to maintain quality and resolve problems. Availability of essential commodities is critical for the delivery of services. Stockouts can be prevented by restocking commodities before complete exhaustion of previous supply by taking into account the consumption rate, maintaining an updated inventory of supplies and ordering them well in time. Resources have to be provided

for this to function. PSBI management needs to be integrated into the regular health management system.

We also recognized that engagement and empowerment of the community are essential to a successful implementation of this strategy that can prove to be very beneficial for the families, which reside in such low resource settings. Mothers and families have to be informed and counselled about the illness signs and when and where to seek appropriate care. Sometimes home visits are unsuccessful because the mother is occupied elsewhere or mobile communication is unavailable to establish a suitable time for the home visit.

Our implementation research had a couple of strengths. First, the LGA administration and the TSU worked together to make this implementation research successful. The objective and the potential benefit to the community was clear and seen as part of LGA's effort for improving the health and survival of infants. The intervention was implemented using existing structures and human resource. Second, the TSU as a scientific partner provided technical support for building capacity, mentoring and building the confidence of health workers and implementing a complex intervention with good quality for sick young infants.

We also identified a few limitations. First, the intensive monitoring and supervision would be difficult to replicate within the routine programme setting. Second, we provided additional medicines and other supplies when needed as the regular health system was unable to provide them. Third, we also provided incentives to the CORPs involved in the study, which ended with the study as the health system doesn't have a mechanism to support this. Fourth, the TSU nurses had to conduct home treatment of sick infants during the strike period by the PHC nurses, which might have contributed to improved outcomes despite the strike. One of the major challenges in Nigeria is to pay regular salary to staff and keeping them motivated.

## Conclusion

We demonstrated that the implementation of the WHO guideline on management of sick young infants with PSBI when a referral is not possible in the Nigerian context is feasible and can contribute to saving many infant lives. However, to scale up this intervention, government commitment will be needed to strengthen the health system, motivate and train health workers, provide necessary commodities, establish technical support for the system and linkages with communities.

## Supporting information

**S1 Fig. PSBI cases identified at PHCs by month.**
(TIF)

**S1 Table. Staff category in each health facility.**
(PDF)

**S1 File. Field supervision checklist.**
(PDF)

**S1 Appendix. Dataset.**
(ZIP)

## Author Contributions

**Conceptualization:** Adejumoke Idowu Ayede, Samira Aboubaker, Shamim Ahmad Qazi, Rajiv Bahl, Adegoke Gbadegesin Falade.

**Data curation:** Oluwakemi Oluwafunmi Ashubu, Kayode Raphael Fowobaje.

**Formal analysis:** Adejumoke Idowu Ayede, Kayode Raphael Fowobaje, Samira Aboubaker, Yasir Bin Nisar, Shamim Ahmad Qazi.

**Funding acquisition:** Adejumoke Idowu Ayede.

**Methodology:** Adejumoke Idowu Ayede, Oluwakemi Oluwafunmi Ashubu, Samira Aboubaker, Yasir Bin Nisar, Shamim Ahmad Qazi, Rajiv Bahl, Adegoke Gbadegesin Falade.

**Project administration:** Adejumoke Idowu Ayede, Adegoke Gbadegesin Falade.

**Supervision:** Adejumoke Idowu Ayede, Oluwakemi Oluwafunmi Ashubu, Kayode Raphael Fowobaje, Shamim Ahmad Qazi, Adegoke Gbadegesin Falade.

**Validation:** Oluwakemi Oluwafunmi Ashubu, Kayode Raphael Fowobaje.

**Writing – original draft:** Adejumoke Idowu Ayede, Samira Aboubaker, Yasir Bin Nisar, Shamim Ahmad Qazi, Rajiv Bahl.

**Writing – review & editing:** Adejumoke Idowu Ayede, Oluwakemi Oluwafunmi Ashubu, Kayode Raphael Fowobaje, Samira Aboubaker, Yasir Bin Nisar, Shamim Ahmad Qazi, Rajiv Bahl, Adegoke Gbadegesin Falade.

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
