## [Decision Letter · Decision Letter 0]

30 Dec 2020

PONE-D-20-33687

Implementation research findings on management of possible serious bacterial infection in young infants where referral is not possible in the context of existing health system structure in Ibadan, South-west Nigeria

PLOS ONE

Dear Dr. Ayede,

Thank you for submitting your manuscript to PLOS ONE. After careful consideration, we feel that it has merit but does not fully meet PLOS ONE’s publication criteria as it currently stands. Therefore, we invite you to submit a revised version of the manuscript that addresses the points raised during the review process.

We recommend major revisions to your paper. Please respond to the reviewer comments especially addressing issues related to the additional resources, support and inputs provided by the project and the implications for sustainability.

We look forward to receiving your revised manuscript.

Kind regards,

Tanya Doherty, PhD

Academic Editor

PLOS ONE

Journal Requirements:

2. Please include a caption for figure 1.

Reviewers' comments:

Reviewer's Responses to Questions

**Comments to the Author**

1. Is the manuscript technically sound, and do the data support the conclusions?

Reviewer #1: No

Reviewer #2: Partly

2. Has the statistical analysis been performed appropriately and rigorously? 

Reviewer #1: Yes

Reviewer #2: No

3. Have the authors made all data underlying the findings in their manuscript fully available?

Reviewer #1: No

Reviewer #2: Yes

4. Is the manuscript presented in an intelligible fashion and written in standard English?

Reviewer #1: No

Reviewer #2: No

5. Review Comments to the Author

Reviewer #1: Thank you for the opportunity to review this manuscript, which reports the implementation research findings on PSBI (management of possible serious bacterial infection in young infants where referral is not possible) from Ibadan, South-west Nigeria. Research, such as this, is critical for understanding how to initiate and implement PSBI interventions in different contexts. I applaud the authors for their work and believe the topic is important. However, the paper, in its current form, is not at standard for publication, and therefore my decision is to reject the paper. My reasons are justified below.

MAJOR ISSUES

1. The authors argue that this study demonstrated implementation “using existing structures and human resources” (page 25, line 450). However, a number of factors were established or implemented for PSBI that seem above and beyond the current capacity of the health system. First, the Technical Support Unit was established for the research project, as explained (page 11). Was the TSU set up to continue after the project ended or did these end with the project? Second, the TSU included project supervisors and project nurses, which one assumes were hired for the project itself. Were these new positions funded by the State MOH with budget lines for these positions to be retained at the end of the projects or were these positions funded by the projects and then positions terminated at the end of the study. Third, the authors note the stipends for the CORPS as a limitation. The assumption is that these stipends ended after the project concluded and the CORPS no longer undertook these responsibilities. More details are required on this point as well. Four, you established a separate monitoring and supervision system, which would not be feasible to implement in the current system. Finally, the commodities were provided by the project, as noted in the limitations section. However, did families still have to pay for these (as they did for other medicines at the PHCs)? Also why were there stockouts when the project was providing these (Table 3)? The authors should reassess their framing that this intervention was implemented through the existing system given it does not seem sustainable beyond the life of the project.

2. There are a couple of issues with the methods section.

a) The data collection process is unclear. In the limitation section, you note the intensive monitoring and supervision undertaken (which would be difficult to replicate). The methods section provides an explanation of the oversight process including the checklist used in supplementary files. But it is not clear as to who was supervising the data collection (this information is not specified on page 14). Also more information is needed on how the case data was collected (e.g. births, identification of suspected cases, treatment adherence, follow up) – who collected it, how was it captured, etc… The assumption is that the CORPS collected this data using the CRFs, but that is never stated or explained.

b) The details provided on data collected and analysis of the implementation factors (barriers and facilitating) is weak and needs further explanation. Who collected the data? Were standard tools developed and used for observations, interactions and events? The fact that there were no formal interviews or focus groups is a limitation. The subjectivity of these findings also needs to be noted as a limitation in the discussion section.

c) The methods section was hard to follow. The sister study paper on Zaria, Wammanda et al 2020 PLoS One, developed a helpful graphic which you may want to adapt for restructuring the methods section.

3. It would be helpful to readers unfamiliar with iCCM and IMCI to better understand these programs and how the PSBI intervention is related (or not) to them. Currently these are mentioned with references but given these are existing platforms from which the intervention is building upon, they warrant description.

4. Linked to the results section:

a) Figure 1 is blurry. These results should be discussed and additional data shown by month (even as a supplementary file). Why was there an increase in the second period?

b) Table 2 provides the results of clinical treatment failure excluding deaths. This term should be defined and explained.

c) Page 20/Table 3 presents the challenges and solutions but neglects to consistently specify the timeframe of these challenges, actions and outcomes. For example, you note that the problem on time was resolved by the third month of the project, but you do not indicate other timeframes for the other challenges identified.

d) Panel 2 indicates treatment would be provided at the PHC on output bases. Yet Table 3 indicates that sick infants were seen at home by TSU nurses during the strike period. This should be mentioned in the limitations section. It was also a learning opportunity to see how home delivery of the service by TSU nurses on how this may or may not have impacted the results – which should be discussed.

e) The results should provide information on the implementation experience. For example, you have included information about how long it took to establish the program in the discussion section (page 24 lines 429-430); yet given you have included the process of establishing the project in the methodology, this information should come in the results section.

5. Discussion section

a) The site for the other study, Kaduna, had a free MNCH policy whereas you have indicated that the cost of treatment is mainly out of pocket in Oyo State. Did that play a role in the comparative outcomes? Were any lessons learned regarding this?

b) How do your findings link to iCCM and IMCI implementation in Oyo State or in other contexts in Nigeria? And in the other literature?

c) Were there referral issues identified (lack of petrol for the ambulances)?

6. Overall the paper has many grammatical errors and requires a serious copy edit.

MINOR ISSUES

- Under study site characteristics, add mortality and related coverage indicators to give a sense of situation.

- Page 6, Line 127, please name the four wards selected.

- Page 7, Lines 146-7: What is the capacity of these referral hospitals e.g. is there oxygen? Adequate space? Family centered care (e.g. space for parents)? This information would be useful given it may be factors linked to parental decision to accept referral or not.

- Page 11, Line 181: Provide more details on the meetings organized with the community leaders (how many and when)?

Reviewer #2: Abstract: the introduction needs to have a few modifications made to ensure that it reads more easily. The English in the abstract is not of the same quality compared to the rest of the paper.

Introduction:

• 2.5 million deaths ….this is globally and should be stated.

• starts with “neonatal” infections. However, later in the introduction it mentions about “young infants up to 2 months of age”. The definition of “neonatal” is less than 1 month of age.

Methods: line 115…enabling factors….

Implementation of study:

• Panel 2: font is different size

Results (295):

• 303…assuming 30% births were missed….what evidence is this 30% based on?

• 304….approximately 45% cases…..what evidence supports this statement?

• Table 1: Infants referred to hospital from PHC: it is stated in column 2,3,4 that 100% of these patients were referred to the hospital. However, the text suggests that the column 2 should be 21/338, column 3 63/462 and column 4 17/22.

Discussion:

• 458 in conclusion…there should be a separate heading of “Conclusion”

6. PLOS authors have the option to publish the peer review history of their article (what does this mean?). If published, this will include your full peer review and any attached files.

Reviewer #1: No

Reviewer #2: No

---

## [Author Response · Author response to Decision Letter 0]

23 Feb 2021

Our responses are in the rebuttal letter.

---

## [Editor Report · Decision Letter 1]

4 Mar 2021

Management of possible serious bacterial infection in young infants where referral is not possible in the context of existing health system structure in Ibadan, South-west Nigeria

PONE-D-20-33687R1

Dear Dr. Ayede,

We’re pleased to inform you that your manuscript has been judged scientifically suitable for publication and will be formally accepted for publication once it meets all outstanding technical requirements.

Kind regards,

Tanya Doherty, PhD

Academic Editor

PLOS ONE
---

## [Editor Report · Acceptance letter]

18 Mar 2021

PONE-D-20-33687R1 

Management of possible serious bacterial infection in young infants where referral is not possible in the context of existing health system structure in Ibadan, South-west Nigeria 

Dear Dr. Ayede:

I'm pleased to inform you that your manuscript has been deemed suitable for publication in PLOS ONE. Congratulations! Your manuscript is now with our production department. 

Kind regards, 

on behalf of

Professor Tanya Doherty 

Academic Editor

PLOS ONE